# Novel Optical Methodology Unveils the Impact of a Polymeric Pour-Point Depressant on the Phase Morphology of Waxy Crude Oils

**DOI:** 10.3390/polym16131933

**Published:** 2024-07-06

**Authors:** Irene Perna, Rosalia Ferraro, Consiglia Carillo, Salvatore Coppola, Sergio Caserta

**Affiliations:** 1Department of Chemical, Materials and Production Engineering, University of Naples Federico II, P. le V. Tecchio 80, 80125 Naples, Italy; irene.perna@unina.it (I.P.); rosalia.ferraro@unina.it (R.F.); consiglia.carillo@libero.it (C.C.); 2CEINGE Advanced Biotechnologies Franco Salvatore, Via G. Salvatore 436, 80131 Naples, Italy; 3Elastomers Research and Development Centre, Versalis S.p.A. (Eni), I-48100 Ravenna, Italy

**Keywords:** waxy crude oils, polymeric pour point depressants, phase morphology, flow assurance

## Abstract

Crude oil, also known as petroleum, plays a crucial role in global economies, politics, and technological advancements due to its widespread applications in industrial organic chemistry. Despite environmental concerns, the dwindling supply of easily accessible oil reservoirs necessitates the exploration of unconventional resources, such as heavy and extra-heavy oils. These oils, characterized by high viscosity and complex composition, pose challenges in extraction, transportation, and refinement. With decreasing temperatures, heavy oils undergo phase changes, with transitions from Newtonian to non-Newtonian fluid behavior, leading to difficulties in transportation. Alternative methods, such as the use of polymeric pour-point depressants, help mitigate flowability issues by preventing wax precipitation. Understanding the properties of waxy crude oil, such as the wax appearance temperature (WAT), is crucial for effective mitigation strategies. The objective of this research is to determine the WATs of different types of waxy crude oils through a comparative analysis using advanced techniques such as cross-polar microscopy (CPM), standard rheology, and differential scanning calorimetry (DSC). Disparities in WAT identified through different analytical methods highlight the potential of microscopy to enhance our understanding of complex fluid dynamics in real time in order to proactively identify and address crystallization issues in oilfields.

## 1. Introduction

Petroleum, commonly referred to as crude oil, represents an indispensable resource, playing a pivotal role in global economies, politics, and technological advancements. Its significance has notably increased due to its widespread applications in industrial organic chemistry, including the synthesis of plastics, fertilizers, solvents, adhesives, and pesticides [1]. Despite concerns regarding its environmental impact, projections indicate that petroleum and its derivatives will continue to satisfy a significant portion, up to 80%, of the world’s energy requirements in the foreseeable future [2,3,4].

However, the depletion of easily accessible and economically viable oil sources necessitates the exploration of unconventional resources [5]. These resources, such as heavy and extra-heavy oil [6,7], are characterized by a high proportion of long-chain linear alkanes (20–40 atoms) [8], which poses unique challenges due to their complex composition and high viscosity. Indeed, crude oil is a complex mixture of hydrocarbons with varying molecular weights, along with other organic compounds [9]. In general, the elemental composition reveals approximately 84–86% carbon, 10–14% hydrogen, and minor amounts of sulfur, nitrogen, and oxygen [10,11,12].

The high viscosity and complex composition of heavy and extra-heavy oils make their production, transportation, and refinement problematic [2,13,14,15]. As temperatures decrease, the precipitation of high molecular weight paraffins and asphaltenes [16,17,18,19,20] induces a transition from exhibiting Newtonian fluid behavior to displaying complex non-Newtonian fluid characteristics, including thixotropic and shear-thinning behavior with yield stress [21]. This phenomenon is also attributed to crystal growth, the three-dimensional evolution of which has been studied through both linear [22] and nonlinear analysis [23,24].

Heating is a common method utilized to overcome problems when transporting heavy oil by pipeline [2,13,25,26,27], but this introduces high capital and operational costs over long distances and faces challenges related to the cooling effect of surrounding water and the maintenance of pumping and heating stations [2,27,28,29,30].

Another alternative for mitigating flowability issues is the use of polymeric pour-point depressants (PPDs) [31,32], also known as wax crystal modifiers. These chemical additives affect nucleation, adsorption, or solubility, thereby reducing wax precipitation. Typically, a PPD consists of an oil-soluble long-chain alkyl group and a polar structure within its molecular composition. The long-chain alkyl group can integrate into the wax crystal in the gas oil, while the polar moiety remains on the wax crystal’s surface. This process inhibits the formation of a crystal lattice and reduces the size of the wax crystals [33].

Understanding the properties of waxy crude oil, such as the wax appearance temperature (WAT), is crucial for implementing effective mitigation strategies [34]. Advanced techniques such as cross-polar microscopy (CPM), standard rheology, and differential scanning calorimetry (DSC) are commonly used to determine the WAT [35,36]. In this study, we conducted a comparative analysis of three distinct types of waxy crude oils, denoted as I, II, and III. These oils vary in their extraction sources, paraffin concentrations, and the presence of PPDs. Our goal was to determine the WAT of each sample through rigorous experimentation by employing the aforementioned methodologies while ensuring the same thermal history across all methodologies employed. Subsequently, we conducted a comparative analysis of the WAT results obtained from these techniques.

Combining various approaches, this study seeks to better understand crystallization phenomena and how to address flow assurance issues. Specifically, the aim is to propose using microscopy as a novel and potentially more sensitive method for investigating crystallization phenomena, comparing it to the standard techniques of DSC and rheology. Indeed, while rheological analysis effectively identifies crystallization, its capability for real-time detection is constrained. Initially, when crystal concentration is low, rheological properties remain largely unchanged. However, as the crystal content increases, discernible alterations in rheological behavior emerge, allowing the crystallization phenomenon to be detected once it is already occurring.

By integrating these diverse analytical approaches, our study not only advances our understanding of waxy crude oil behavior but also highlights the potential for microscopy to enhance our ability to probe complex fluid dynamics in real time.

## 2. Materials and Methods

### 2.1. Sample

Three distinct crude oils (COs), referred to as CO I, CO II, and CO III, were examined in this study. CO II and CO III were extracted at different time points from the same oil field, with CO II being extracted earlier than CO III. CO I, which was the lightest among the three due to its earlier extraction, served as the control sample.

Additionally, CO II underwent treatment with two distinct PPDs, labeled ADD-I and ADD-II. This approach allowed the examination of differences in the physical attributes and chemical characteristics among these crude oils, as well as an assessment of the impacts of specific additives on their performance. Consequently, CO I was examined at three different concentrations (5%, 10%, and 20% *w*/*w*) of paraffin (Paraffin Wax from Fluka Analytical) as a reference. These three concentrations of paraffin were chosen with the aim of creating model systems with significant variations between samples. This ensures that the observed differences are much greater than those typically seen between samples from the same oil field. The details of the investigated crude oils are summarized in Table 1. Samples were prepared in a fume cupboard to comply with safety protocols necessitated by the toxicity of crude oil. Upon the initial measurement of CO, the appropriate quantity of paraffin wax was introduced. Subsequently, the samples were mixed and heated until the complete dissolution of the paraffin wax.

### 2.2. Thermal Profiling

The thermal profile consisted of several phases, as depicted in Figure 1, beginning with a cooling phase from 30 °C to 4 °C (points 1 to 2), followed by a 10 min period of maintaining a constant temperature at 4 °C (points 2 to 3). Subsequently, the temperature was raised from 4 °C to 50 °C (points 3 to 4), followed by another cooling cycle from 50 °C back to 4 °C (points 4 to 5). Another 10 min interval of constant temperature at 4 °C ensued (points 5 to 6), concluding with a heating phase from 4 °C to 50 °C (points 6 to 7). The heating and cooling rate was set to 1 °C per minute, which was chosen based on typical values used for crystallization studies [37,38]. Specifically, we selected rates that are not excessively high to ensure the accurate determination of characteristic temperatures. In DSC, very low ramp rates can lead to weak signals and a poor signal-to-noise ratio. In rheology, extremely low ramp rates increase the risk of evaporating the volatile fraction. The maximum temperature of 50 °C reached during the cycles is a balance between the need to anneal the paraffins and the minimization of the evaporation of the volatile fraction. This standardized thermal history was systematically applied to all tests conducted using various techniques to facilitate a comprehensive comparison of the obtained results.

### 2.3. Differential Scanning Calorimetry

Crystallization was investigated using a standard method employing differential scanning calorimetry (DSC), specifically the Q1000 model from TA Instruments, which was equipped with a liquid nitrogen cooling system (LNCS). This system offers exceptional performance and flexibility in cooling, with capabilities ranging from an operational low of −180 °C to a high-temperature limit of 550 °C and a rapid cooling rate capacity of up to 140 °C/min. The DSC furnace houses two identical pans made from a material that withstands test temperatures without interacting with the sample. Approximately 3 mg of the sample was loaded into one pan, hermetically sealed using a manual press, and then placed on thermoelectric disks inside the furnace. The other pan, kept empty, served as a reference for differential measurement.

The furnace is hermetically sealed via the control unit to isolate the test environment. After programming the thermal analysis, an inert atmosphere, typically Ar or N_2_, is introduced to the furnace, ensuring a continuous and uniform flow around the material under analysis.

DSC analysis hinges on the evaluation of the latent heat of fusion released during crystallization, identifiable as a peak in the thermogram upon cooling. Both the sample and reference pans are maintained at nearly identical temperatures throughout the experiment. Any temperature variation between them is attributable to physical changes in the sample, such as phase transitions. Depending on whether the process is endothermic or exothermic, varying amounts of heat will be required to maintain the sample and reference pans at the same temperature.

Throughout the experiment, the DSC instrument measured the temperature difference between the sample and reference pans using thermocouples, estimating the thermal flux of the reactions occurring within the material.

### 2.4. Rheological Setup

All rheological measurements were performed within a controlled-stress rheometer (MCR 301, Anton Paar Italia S.r.l, Rivoli (Italy)) equipped with a dual Peltier system to ensure precise temperature regulation. A coaxial cylinder geometry with a diameter of 17 mm was utilized to amplify torque signal intensity and address concerns related to the evaporation of the sample (∼5 mL). To minimize sample evaporation, a custom-made cap was placed on the top. To effectively saturate the surrounding environment and reduce sample loss, a cotton–wool disk soaked in crude oil was attached to the inside of the cap.

Temperature sweep tests were carried out under static conditions to investigate complex viscosity, η*, and both storage and loss moduli, G′ and G′, respectively. The samples were initially cooled to an initial temperature of 30 °C, and, subsequently, oscillatory measurements were performed, applying the same thermal history reported and described in Figure 1. A low deformation of 0.03% was maintained throughout the tests to ensure static conditions and mitigate potential slippage concerns.

Additionally, since the volume and density of the samples fluctuated during the test, thereby exerting increasing force on the rheometer’s axle, a zero normal force (F_N_ = 0) was set to prevent potential engine damage and measurement alterations. The normal force transducer was employed to detect variations in force, facilitating automatic adjustment of the gap. This adaptation guaranteed the consistency of the normal force and measurement accuracy.

### 2.5. Cross-Polar Microscopy

Cross-polar microscopy (CPM) is employed to examine the dynamic evolution of microcrystalline structures. This technique is particularly well suited for examining wax crystallization due to the optical anisotropy present in non-cubically structured crystalline materials, leading to birefringence. Consequently, wax crystals rotate the plane of polarized light, rendering them luminous against the darker background of liquid oil or amorphous crystals when observed under polarized light.

CPM experiments were conducted by means of an optical inverted microscope (Axiovert 200, Carl Zeiss S.p.A, Milan (Italy)) equipped with a 20× magnification and polarized microscopy. The microscope was situated on an anti-vibration table and featured a focusing mechanism and a motorized stage capable of precise and automated adjustments within the sample’s field of view. Additionally, the microscope stage incorporated an incubator with a water circulation system connected to a water bath to ensure precise temperature regulation within the incubator. This setup facilitated the imposition of specific thermal profiles on the samples.

Approximately 60 μL of samples are placed in a micro-pool realized on a microscope slide using 150 µm spacers with double-sided tape, with a coverslip glass on the top to prevent evaporation. Images are captured using a CCD monochrome camera (Hamamatsu 1394 ORCA-ER CCD camera) and managed with an Objective Imaging controller. This system was operated via the custom-made Time-Lapse software within the LabVIEW environment, ensuring consistent and periodic scanning of predetermined sample areas. Specifically, four images were captured for each sample at the center of the micro-pool at intervals of approximately 3 min, spanning a total duration of 10 h.

### 2.6. Image Analysis

To quantify the temperature’s impact (T) on the structure of each sample and track the evolution of crystallization phenomena, images were analyzed using the Image Pro Plus 4.5 software. Four images were captured for each sample at intervals of approximately 3 min, covering a total test duration of 10 h. For each image, the mean gray level (I) was calculated and examined. In detail, specific regions of interest (ROIs) were selected within each image to focus on the area most relevant to our analysis. The software was then utilized to calculate the mean gray level within the selected ROIs. This process involved converting the images into grayscale and analyzing the pixel intensity values within each ROI. To facilitate comparative analysis over time and temperature, I was normalized with respect to the mean gray value of the image at the initial time (I_0_) and plotted against T.

### 2.7. Statistical Analysis

Data in this study are presented as the mean ± standard error of the mean (SEM). The SEM is a statistical tool that quantifies the amount of variability one might expect in a sample mean relative to the population mean, thereby facilitating statistical inference from the sampling distribution. It is calculated by dividing the standard deviation by the square root of the sample size.

## 3. Results

In this section, we elucidate the outcomes obtained from CPM, standard rheology, and DSC to determine the WATs of three varieties of waxy crude oils. This investigation considers extraction origins, paraffin content, and the inclusion of PPDs.

### 3.1. Cumulative Heat Flow Analysis

The cumulative heat flow values for samples B, C, D, and E, as depicted in Figure 2, were derived from the heat flow (W/g) vs. temperature (°C) plot provided in the Appendix A. The cooling phase is depicted by the blue lines, while the heating phase is represented by the red lines. Arrows with the same color code are added to help with the visualization of the thermal cycle. We aimed to demonstrate the appropriateness of the chosen annealing temperature by conducting a complete thermal cycle on the model sample with the highest paraffin content (sample D) in order to validate our thermal profile. For the other samples, our primary focus was on characterizing their melting behaviors under realistic conditions following prolonged exposure to ambient temperatures. For this reason, we decided to conduct the experiment only on the reported samples by applying the first thermal cycle.

The cumulative value is assessed with the consideration that the baseline of the heat flux signal, representing the constant heat flow value before and after crystallization, should ideally be zero. Thus, the area under the peak of the heat flow can be directly linked to the latent heat of crystallization. Consequently, the cumulative value, which integrates this area only up to a specific temperature where crystallization is incomplete, serves as a measure of the enthalpy of crystallization of the sample. Moreover, this quantity of heat released during crystallization can be associated with the amount of wax crystallized [39]. The temperature at which the cumulative value decreases can be used to evaluate the WAT. The complete data of the cumulative heat flow of both cycles are available only for sample D.

Regarding sample B (Figure 2a), during the initial cooling ramp (1–3), the decrease in the cumulative heat flow is more pronounced than the increase in cumulative heat flow associated with the subsequent heating ramp (3–4). Indeed, the latent heat of crystallization, H_C_, is 3.37 J/g, whereas the latent heat of melting, H_m_, is 2.28 J/g, which is 67% lower.

The opposite situation is observed in samples C and D (Figure 2b and c, respectively), as they are partially crystallized at the beginning of the test. Indeed, the plateau observed at point 4 (end of the heating process) occurs at a higher value of the cumulative heat flow with respect to the one in point 1 (beginning of the cooling process). Regarding sample D, which has a higher paraffin concentration (20% *w*/*w*), by comparing the first and the second cycles (1–3 and 4–6, respectively), the second cooling ramp leads to a higher heat release with respect to the first, which is probably associated with the partially crystallized structure at point 1. In fact, H_C1_, associated with the first cooling, is 2.44 J/g, while H_C2_ is 25.53 J/g.

As regards sample E (Figure 2d), the heating line (3–4) shows a different development in contrast to the other samples, with a decreasing trend, suggesting that the crystallization was not complete and, thus, continued during the heating. Furthermore, no evidence of the melting process during the heating ramp was detected, which was most probably because of the low heat flow signal.

### 3.2. Rheological Characterization

Temperature sweep tests were conducted to analyze the evolution of G′, G″, and η* with respect to the temperature for each sample of the crude oils investigated in this study, also highlighting the effect of paraffin concentration (samples B, C, and D), extraction time (samples E and F), and PPDs (samples E, G, and H).

In Figure 3, the G′ values during the entire thermal profile are presented, whereas G″ and η* are reported in the Appendix A. Downward triangle symbols represent cooling, while upward triangle symbols represent heating, distinguished by filled and empty symbols for the first and second cycles, respectively. Data related to the 10 min hold at constant temperature (4 °C) are indicated by circles and squares. Insets providing a closer view of the data are available in each plot. It is noteworthy that the second temperature cycle (4–5) yields higher modulus values than the first cycle (1–2), which is attributed to the 10 min hold at 4 °C (2–3 and 5–6), as is evident in the evolution of G′ during the sample’s exposure to 4 °C.

For all samples, G′ decreases significantly with temperature by at least four orders of magnitude. For instance, as the temperature increases from 4 °C to approximately 20 °C, G′ decreases from approximately 10^5^ to 10^1^, reaching a plateau thereafter.

Figure 3a illustrates the effect of the percentage of paraffin in terms of G′ during the thermal analysis. As the percentage of paraffin increases, G′ also increases, and the WAT shifts to higher values. Additionally, sample D displays a different behavior from that of the other samples at low temperatures, possibly due to solid-sample slippage within the measuring geometry, even imposing a lower deformation.

Regarding the effect of the extraction time in Figure 3b, a significant observation arises when comparing the G′ curve of sample E with that of sample F, which were extracted earlier and later, respectively. Notably, they overlap during the second cycle (4–5 and 6–7), whereas in the first cycle, the G′ curves of sample F are shifted upward with respect to the G′ curves of sample E, which is probably related to a preexisting crystal structure in the sample before the first heating cycle.

In terms of the type of PPDs added to sample E (ADD I and ADD II, resulting in samples G and H, respectively), ADD I reduces the WAT and G′ of crude oil, as shown in Figure 3c, indicating a stronger microscopic interaction.

### 3.3. Cross-Polar Microscopy

#### 3.3.1. Phase-Contrast Microscopy Insights

Representative phase-contrast microscopy images acquired with crossed polarizers are presented in Figure 4. The effect of temperature is reported in the columns, with each temperature ranging from 4 °C to 50 °C, as denoted by the corresponding phase number during the thermal profile (see Section 2.2). Each investigated sample, characterized by a distinct concentration of paraffin or PPDs and extraction time, is labeled from A to H following the nomenclature outlined in Table 1 and reported in the rows. It is noteworthy that images 2 and 5 were omitted due to overlap with the images depicting conditions 3 (=2) and 6 (=5). Additionally, images of sample A are not included, as crystallization did not occur throughout the entire test.

Regarding the first three samples reported in the panel, it is observed that the number of crystals increases with higher paraffin concentrations, from 5%*_w_*_/*w*_ to 20%*_w_*_/*w*_. Additionally, regardless of concentration, crystal proliferation is noted as temperatures decrease, followed by gradual dissolution upon heating (from points 4 to 7). During the initial temperature ramp (1–2), crystal formation is influenced by existing structures, resulting in new paraffin aggregating to them, thus forming larger crystals. Conversely, crystals formed during the second temperature ramp (3–4) are either completely or partially dissolved at the outset, leading to nucleation processes and the formation of smaller crystals.

In contrast, samples E and F, corresponding to crude oils extracted at two different times, exhibit a more uniform appearance and do not show crystal formation after heating to 50 °C. Instead, they display black structures, likely representing an amorphous solid phase.

Crude oils G and H, which contain two different PPDs, behave differently from virgin crude oil E. Crystallization is observed after the cooling process, indicating that the presence of PPDs has a detrimental effect on the sample.

#### 3.3.2. Variation in Light Intensity with Temperature

Figure 5 illustrates the variation in the mean intensity of light (I), normalized to the value at time zero (I_0_), with temperature for each sample, providing a quantitative assessment of the observations made in Figure 4. Downward triangle symbols are indicative of cooling, while upward triangle symbols are utilized to represent heating. The two cycles are further distinguished by filled symbols for the first cycle and empty symbols for the second cycle. Arrows with the same color code are added to help with the visualization of the thermal cycle.

I/I_0_ of sample A remains roughly constant during the whole experiment, showing a maximum value of 1.19 and a minimum of 0.98, which is indicative of the absence of the crystallization phenomenon.

Samples B, C, and D show an increase in I/I_0_ as T decreases both in the first and second cycle, passing from a morphology characterized by a high birefringence due to the presence of the waxy crystals to an amorphous structure.

However, the curves of samples E, G, and H do not exhibit an increase in brightness, which remains approximately constant during the second cooling, due to the presence of black solid structures that influenced I/I_0_ and affected the measurement of the WAT for these crude oils.

### 3.4. WAT Evaluation

A WAT evaluation is conducted for each approach employed here (DSC, rheology, and CPM), encompassing both cycles, namely, the first and the second (referred to as WAT_1_ and WAT_2_, respectively). It is imperative to acknowledge that the WAT data concerning the first cycle cannot be assessed as the true WAT values due to the occurrence of crystal formation in the early stages of the process. Despite this, we still decided to report the data from the first thermal cycle, as it is interesting to study the behavior of the sample following melting in “realistic” conditions, i.e., after a long stay at T_amb_. The results of both cycles are reported for numerical comparison in Figure 6a (WAT_1_) and Figure 6b (WAT_2_).

In the case of DSC, the WAT is evaluated as the temperature corresponding to a decrease in the cumulative heat flow. Concerning rheology, the WAT is assessed as the temperature corresponding to an increase in G′, consequently leading to a change in the slope of the trend. For CPM, the WAT can be determined as the temperature at which the intensity, I/I_0_, increases during the cooling process. An example of the calculation of the WAT using all three experimental methodologies is provided for sample D in the Appendix A.

For all approaches, sample A exhibits no crystallization phenomena, thus impeding the identification of its WAT. Consistently with the microscopic images, which reveal an increased presence of waxy crystals with higher paraffin concentrations, the WAT of samples B, C, and D shifts to higher T values during the second cycle. For example, for sample B, the WAT_1_ associated with the first cooling is about 27 °C, whereas the WAT_2_ identified for the second cycle is roughly 32 °C. For samples E, G, and H it was not possible to precisely evaluate the WAT from the optical analysis, which was in agreement with the microscopic images, due to the presence of black solid structures. This structural characteristic significantly impacted the level of I/I_0_ and consequently affected the accuracy of the measurement.

As depicted in Figure 6a,b, assuming the reliability of the DSC evaluation, the temperatures identified from the rheological data appear to be lower, which is attributable to the technique’s limitation of viscoelastic properties only being affected when the amount of wax is significant, thus detecting the crystallization phenomenon only after it has already begun. In contrast, a closer alignment is observed between the outcomes of DSC and microscopy, suggesting a predictive advantage of the microscopic approach. Indeed, for sample D, the WATs detected through DSC and microscopy are 42 °C and 45.5 °C, respectively, while, from the rheological data, a WAT of 26.5 °C was obtained. The same considerations can be made for sample B in the WAT data for the first cycle. Moreover, the WAT detected through microscopy appears to be higher than that from the rheological data, indicating that microscopy may offer a more predictive technique for studying the phenomenon.

It is noteworthy that each technique yields distinct hysteresis loops. Specifically, the hysteresis cycle of the rheological data depicts the heating curve (red line) succeeding the cooling curve (blue lines), conversely to the hysteresis cycle that emerged from the microscopic analysis.

## 4. Discussion

In this study, the crystallization phenomena of waxy crude oils are investigated due to their persistent challenge for the petrochemical industry [2,13,14,15]. A thorough comprehension of this intricate issue is fundamental to devising optimal removal strategies [34]. Heating, a common method utilized to overcome issues in the transportation of heavy oil via pipelines [2,13,25,26,27], introduces high capital and operational costs over long distances and faces challenges related to the cooling effect of surrounding water and the maintenance of pumping and heating stations. Another alternative for mitigating flowability issues is the use of polymeric PPDs, also known as wax crystal modifiers. Understanding the properties of waxy crude oil, such as the WAT, is crucial for implementing effective mitigation strategies.

In the present investigation, a comparative examination of three varieties of waxy crude oils is conducted, considering extraction origins, paraffin content, and the inclusion of PPDs. CPM, standard rheology, and DSC [35,36] are employed to determine the WAT.

From the comparative assessments of crystallization temperatures, a discrepancy can be observed between DSC and rheology, which identify the appearance of wax only when the amount of wax is significant, underscoring the importance of considering multiple methodologies and their respective findings in comprehensive analyses. Conversely, the predictive efficacy of microscopy over other methods emerged, suggesting its viability for implementing real-time control systems.

## 5. Conclusions

In this work, the crystallization phenomena of waxy crude oils were investigated since they are a persistent challenge for the petrochemical industry. A thorough comprehension of this intricate issue is fundamental to devising optimal removal strategies. To this end, this study augmented conventional rheological and calorimetric methods with microscopic analyses to elucidate the process of crystal formation. Specifically, in the present investigation, we conducted a comparative examination of three varieties of waxy crude oils, identified as crude oil I, II, and III. Their extraction origins, paraffin content, and inclusion of polymeric PPDs were examined during the study, as they are variables controlling a crude oil’s characteristics and may alter its behavior. Our primary objective was to ascertain the WAT of each sample through meticulous experimentation utilizing rheology, DSC, and microscopy analysis. Through the comparison of the data provided by these diverse techniques, each offering unique perspectives on the same phenomenon, a comprehensive characterization of crude oil crystallization was achieved. These discrepancies underscore the importance of considering multiple methodologies and their respective findings in comprehensive analyses. Notably, from the comparative assessments of crystallization temperatures, the predictive efficacy of microscopy over other methods emerged, suggesting its viability for implementing real-time control systems. Indeed, as evidenced by our previously discussed results, CPM allows for the detection of incipient crystallization phenomena at temperatures higher than those required to measure relevant changes in rheological parameters. In other words, visual inspection can predict the risk of flow blockage well before any significant rheological change affects the fluid. Therefore, such systems hold the potential to identify and mitigate crystallization issues in oilfields in advance, allowing for timely intervention before alterations in crude oil rheological behavior occur. Indeed, when approaching the WAT, the rheological properties are changed, as the rheological parameters increase by several orders of magnitude in the frame of a few degrees, with the risk of inducing catastrophic flow blockage. Nevertheless, rheology offers critical insights into the flow behavior and viscosity of crude oil, which is vital for engineering processes.

A further advantage of the optical analysis proposed here is related to the possibility of it being implemented with equipment of limited complexity and costs, such as a smartphone camera coupled with polarized filters. This approach can allow the identification of changes in structure at a macroscopic level through visual inspection in a significantly shorter timeframe and in a simpler way compared to the more precise, but more sophisticated, investigations run in labs using rheometers, DSC, or microscopes. The methodology proposed in this work can be used to design simple instruments that can be integrated directly into a processing plant to obtain in-line measurements able to predict phase changes before dramatic variations in rheological properties induce potentially catastrophic events, such as blockages of pipes, enabling prompt precautionary measures.

## Figures and Tables

**Figure 1 polymers-16-01933-f001:**
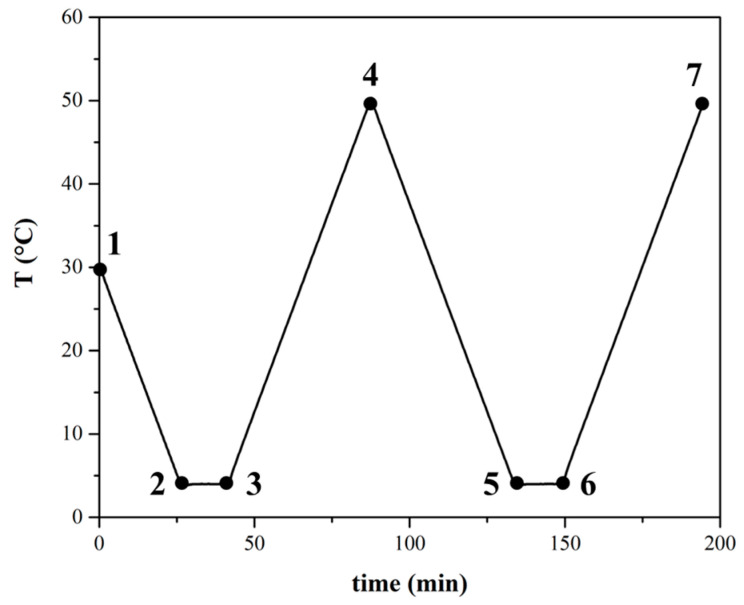
The imposed thermal profile initiates with a rapid cooling phase from 30 °C to 4 °C (points 1–2), followed by a 10 min isothermal hold at 4 °C (points 2–3). Subsequently, a controlled heating phase from 4 °C to 50 °C (points 3–4) is executed, succeeded by a controlled cooling phase from 50 °C back to 4 °C (points 4–5). Another 10 min isothermal hold at 4 °C follows (points 5–6) before the profile concludes with a second controlled heating phase from 4 °C to 50 °C (points 6–7). The heating and cooling rates were maintained at a constant 1 °C per minute.

**Figure 2 polymers-16-01933-f002:**
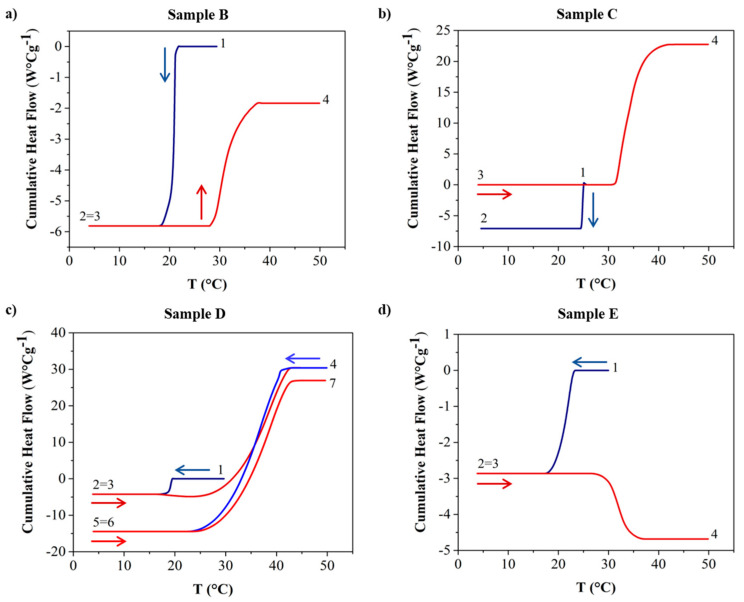
Cumulative heat flow versus temperature, T, for samples B, C, D, and E in (**a**), (**b**), (**c**), and (**d**), respectively. The imposed thermal profile follows the following previously defined code: points 1–2 for the first cooling ramp, points 2–3 for a 10 min holding period at a specified temperature, points 3–4 for the first heating ramp, points 4–5 for the second cooling ramp, points 5–6 for another 10 min holding period at 4 °C, and points 6–7 for the second heating ramp. The red lines indicate the heating phase, while the blue lines denote the cooling phase of the imposed thermal profile. Arrows with the same color code are added to help visualize the thermal cycle.

**Figure 3 polymers-16-01933-f003:**
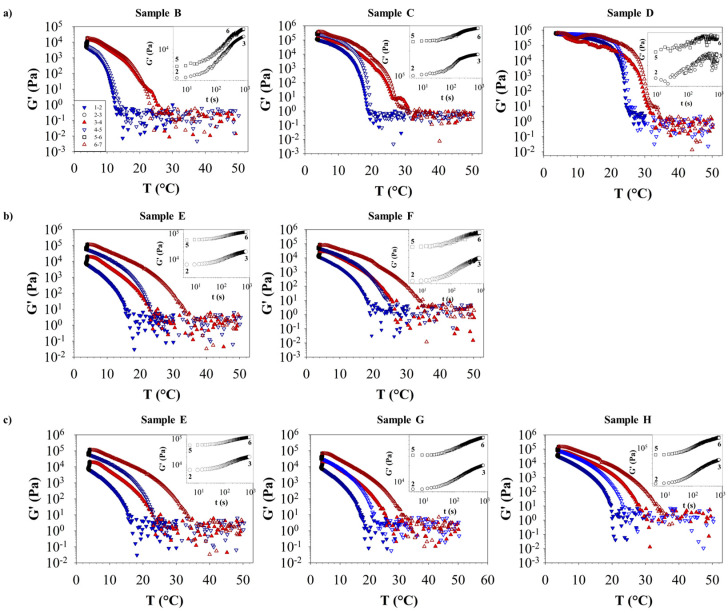
Elastic modulus, G′, versus temperature, T, highlighting the effects of paraffin concentration (**a**) for samples B, C, and D, extraction time (**b**) for samples E and F, and PPD percentage (**c**) for samples E, F, and H. The imposed thermal profile follows the following previously defined code: points 12 for the first cooling ramp, points 23 for a 10 min holding period at a specified temperature, points 3–4 for the first heating ramp, points 4–5 for the second cooling ramp, points 5–6 for another 10 min holding period at 4 °C, and points 6–7 for the second heating ramp. Blue downward triangle symbols represent cooling, while red upward triangle symbols represent heating, distinguished by filled and empty symbols for the first and second cycles, respectively.

**Figure 4 polymers-16-01933-f004:**
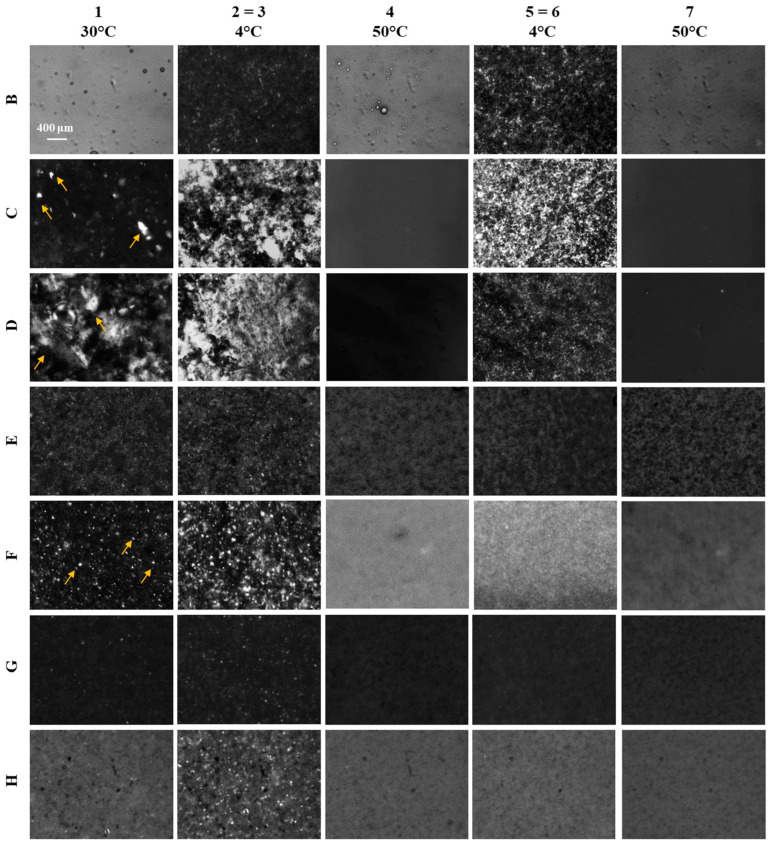
Representative phase-contrast microscopy images of samples B, C, D, E, F, G and H, arranged in rows, captured at different temperatures, arranged in columns, according to the temperature history previously described and reported in Figure 1. Specifically, point 1 defined the beginning of the test at 30 °C, point 2 = 3 involved a 10 min holding temperature of 4 °C, point 4 involved a heating ramp at 50 °C, point 5 = 6 involved a 10 min holding temperature of 4 °C, and point 7 marked the conclusion of the test at 50 °C. Yellow arrows are shown to highlight the crystalline structures.

**Figure 5 polymers-16-01933-f005:**
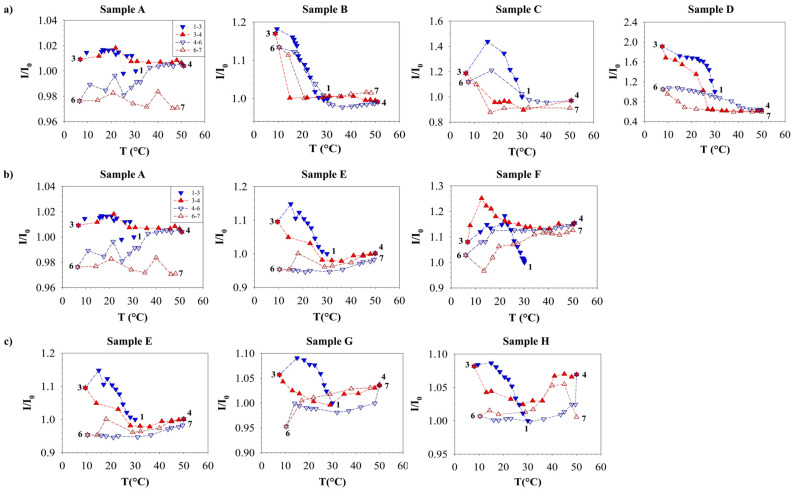
The mean intensity of light (I), normalized with respect to the value at time zero (I_0_) for each sample, versus temperature T. The data allow the identification of the effect of paraffin concentration (**a**) for samples B, C, and D, extraction time (**b**) for samples E and F, and PPD percentage (**c**) for samples E, F, and H. The imposed thermal profile follows the following previously defined code: points 12 for the first cooling ramp, points 23 for a 10 min holding period at a specified temperature, points 34 for the first heating ramp, points 45 for the second cooling ramp, points 56 for another 10 min holding period at 4 °C, and points 67 for the second heating ramp. Blue downward triangle symbols represent cooling, while red upward triangle symbols represent heating, distinguished by filled and empty symbols for the first and second cycle, respectively. The maximum percentage error is 15.38%.

**Figure 6 polymers-16-01933-f006:**
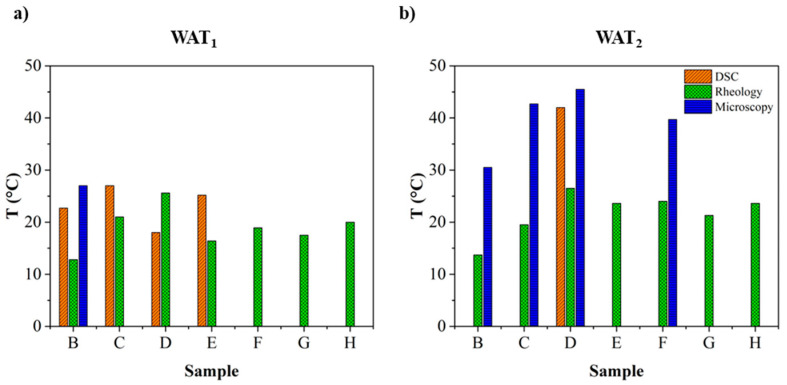
Comparison of the WATs obtained for all samples investigated during the first and second cycles, referred to as WAT_1_ (**a**) and WAT_2_ (**b**), using three approaches: differential scanning calorimetry (DSC), rheology, and contact profilometry (CPM). The maximum percentage error evaluated in the WAT_2_ data with respect to the microscopy results is 5.32%.

**Table 1 polymers-16-01933-t001:** Samples under investigation. Crude oil I served as a model fluid (A). The paraffin effect was investigated by adding 5%, 10%, and 20% *w*/*w* paraffin content to crude oil I (B, C, and D). Crude oil II and crude oil III (E and F) differed in their extraction time. Crude oil II was subjected to treatment with two different PPDs (G and H).

ID Sample	Extraction Time and Concentration Details
A	Crude oil I
B	Crude oil I + 5%*_w_*_/*w*_ of paraffin
C	Crude oil I + 10%*_w_*_/*w*_ of paraffin
D	Crude oil I + 20%*_w_*_/*w*_ of paraffin
E	Crude oil II
F	Crude oil III
G	Crude oil II + ADD-I
H	Crude oil II + ADD-II

## Data Availability

The original contributions presented in the study are included in the article/Appendix A, further inquiries can be directed to the corresponding authors.

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
