# Peer review of "Novel Optical Methodology Unveils the Impact of a Polymeric Pour-Point Depressant on the Phase Morphology of Waxy Crude Oils"

_polymers, 2024, doi:10.3390/polym16131933_

Round 1

Reviewer 1 Report

Comments and Suggestions for Authors

Overall this draft makes a good contribution to the field, by discussing a novel optical methodology. The data collection and analysis part is rigorous, while the discussion and presentation could be improved. I have attached a list of comments and suggestions for revision below, hoping to make the manuscript more accessible to the audience. After moderate revision, the draft should be qualified to be published.

  1. Figure 3, consider enlarging the plots and increasing the resolution a little. Currently, the plots seem somehow blurry.
  2. Figure 4, since crystallization is discussed a lot in the draft, so I would suggest adding some more illustrations to the existing raw images in Figure 4 to emphasize the difference.
  3. Similarly, for Figure 5, consider increasing the resolution.
  4. Figure 6, the color scheme used here might be difficult to read for people with color vision deficiency, consider at least changing the red/green colors.
  5. Also regarding Figure 6, could the error bars be added here?
  6. Line 52, in the introduction and discussion sessions, the authors have made the points that PPD is a better alternative than heating due to lower capital and operational costs. Would it be possible to add some sources to prove this? The sources could be accurate numbers or estimates.
  7. In the materials and methods section, would it be possible to provide vendor information for the crude oil and ADDs?
  8. Line 401, there seems to be a redundant use of the full form of PPD.
  9. Line 406, similar case for CPM.
  10. The discussion section could be enhanced with further discussion over topics including but not limited to, the possibility of using a combination of different tools, and some cost analysis of the methods studied in the draft.

Reviewer 2 Report

Comments and Suggestions for Authors

The research topic is relevant due to the (still) crucial role of oil in the global economy and the need to solve the problems associated with heavy oil, especially in terms of production, transportation and refining. The work demonstrates a modern approach as it uses advanced analytical techniques such as cross-polar microscopy (CPM), standard rheology and differential scanning calorimetry (DSC). The experimental design appears to me to be robust and comprehensive. The scientific novelty of the work is associated with the comparative analysis of the analytical methods used. This allowed the authors to conduct an interesting study of crystallization phenomena and increase the reliability and depth of the results obtained. The authors also acknowledge limitations and suggest areas for improvement, such as improved control of thermal history in microscopy, which indicates a thoughtful and critical approach to experimental design.

The clarity of the article is generally good, with a well-structured presentation of the research background, objectives, methods, and findings. The authors provide detailed descriptions of the experimental procedures and results, facilitating comprehension. However, some sections could benefit from additional clarity, especially in explaining complex concepts to ensure broader accessibility to readers.

 Here are some comments that could be pointed out:

 1 Introduction

The introduction provides a broad overview but could benefit from a more focused discussion on the specific gaps in current research that this study aims to address. It should more clearly articulate how the findings will advance the field. It seems that at the end of the Introduction (last paragraph) you should not write about the work already done (this is appropriate to do in the Conclusion). It is better to clearly formulate the purpose of the study.

  1. Materials and Methods

2.1 While the methodology is detailed, it lacks a clear rationale for selecting the specific percentages of paraffin added to crude oil I. Please explain how these concentrations were chosen?

2.2 The thermal profile used in the experiments is well-documented, but there is no discussion on how the cooling and heating rates were optimized. Please explain how you chose these parameters?

2.3 Line 192. You mentioned the use of Image Pro Plus 4.5 software for image analysis but does not provide sufficient detail on the criteria used for analyzing the mean grey level. Please describe in more detail the stages of image processing.

  1. Results

3.1 You mentioned discrepancies in WAT values ​​obtained using different methods, and this is an important finding of the work. It seems that a more thorough analysis of the reasons for these discrepancies should be carried out. A summary table comparing the WAT results from different techniques for each sample would provide a clearer visual comparison and aid in understanding the differences observed.

3.2 The significance of the findings, especially regarding the practical applications of the novel microscopy technique, needs further elaboration. How can these scenarios be implemented in real practice?

3.3 I would like to see some Figures, especially 3 and 5, in a clearer and larger version.

Conclusion

The conclusion briefly mentions the potential of microscopy but does not fully discuss the broader impact of this research. A more detailed discussion on how this study advances the field and suggestions for future research directions would be beneficial.

 The study appears worthy of publication in a journal Polymers and will be of interest to readers. A response to these small comments would, in my opinion, greatly improve the manuscript, making a clearer, more comprehensive, and impactful contribution to the field.

Reviewer 3 Report

Comments and Suggestions for Authors

Please find comments in the attached pdf. 

Comments on the Quality of English Language

Minor corrections are required. 

Round 2

Reviewer 3 Report

Comments and Suggestions for Authors

The responses provided by the authors to my comments are satisfactory. No further questions or comments from my side.